# Anti-MRSA Constituents from *Ruta chalepensis* (Rutaceae) Grown in Iraq, and *In Silico* Studies on Two of Most Active Compounds, Chalepensin and 6-Hydroxy-rutin 3′,7-Dimethyl ether

**DOI:** 10.3390/molecules26041114

**Published:** 2021-02-19

**Authors:** Shaymaa Al-Majmaie, Lutfun Nahar, M. Mukhlesur Rahman, Sushmita Nath, Priyanka Saha, Anupam Das Talukdar, George P. Sharples, Satyajit D. Sarker

**Affiliations:** 1Centre for Natural Products Discovery, School of Pharmacy and Biomolecular Sciences, Liverpool John Moores University, James Parsons Building, Byrom Street, Liverpool L3 3AF, UK; bio.shaymaa@yahoo.com (S.A.-M.); sushmitanath84@gmail.com (S.N.); G.P.Sharples@ljmu.ac.uk (G.P.S.); 2Laboratory of Growth Regulators, Institute of Experimental Botany ASCR & Palacký University, Šlechtitelů 27, 78371 Olomouc, Czech Republic; 3Medicines Research Group, School of Health, Sport and Bioscience, University of East London, Water Lane, London E15 4LZ, UK; m.rahman@uel.ac.uk; 4Cancer Biology and Inflammatory Disease Division, CSIR-Indian Institute of Chemical Biology, Kolkata, West Bengal 700032, India; priaz12@rediff.com; 5Department of Life Science and Bioinformatics, Assam University, Silchar, Assam 788011, India; adtddt@gmail.com or

**Keywords:** *Ruta chalepensis*, Rutaceae, chalepensin, rutin, anti-MRSA, *in silico*

## Abstract

*Ruta chalepensis* L. (Rutaceae), a perennial herb with wild and cultivated habitats, is well known for its traditional uses as an anti-inflammatory, analgesic, antipyretic agent, and in the treatment of rheumatism, nerve diseases, neuralgia, dropsy, convulsions and mental disorders. The antimicrobial activities of the crude extracts from the fruits, leaves, stem and roots of *R. chalepensis* were initially evaluated against two Gram-positive and two Gram-negative bacterial strains and a strain of the fungus *Candida albicans*. Phytochemical investigation afforded 19 compounds, including alkaloids, coumarins, flavonoid glycosides, a cinnamic acid derivative and a long-chain alkane. These compounds were tested against a panel of methicillin-resistant *Staphylococcus aureus* (MRSA) strains, i.e., ATCC 25923, SA-1199B, XU212, MRSA-274819 and EMRSA-15. The MIC values of the active compounds, chalepin (**9**), chalepensin (**10**), rutamarin (**11**), rutin 3′-methyl ether (**14**), rutin 7,4′-dimethyl ether (**15**), 6-hydroxy-rutin 3′,7-dimethyl ether (**16**) and arborinine (**18**) were in the range of 32–128 µg/mL against the tested MRSA strains. Compounds **10** and **16** were the most active compounds from *R. chalepensis*, and were active against four out of six tested MRSA strains, and *in silico* studies were performed on these compounds. The anti-MRSA activity of compound **16** was comparable to that of the positive control norfloxacin (MICs 32 vs 16 μg/mL, respectively) against the MRSA strain XU212, which is a Kuwaiti hospital isolate that possesses the TetK tetracycline efflux pump. This is the first report on the anti-MRSA property of compounds isolated from *R. chalepensis* and relevant *in silico* studies on the most active compounds.

## 1. Introduction

Antibiotic resistance is a global public health problem and is most common in developing countries [1]. According to the World Health Organization (WHO), every year, microbial resistance to antibiotics causes more than 60,000 deaths worldwide, out of which 77% are children [2]. Microorganisms, particularly bacteria, develop resistance to antimicrobial drugs, mainly because of clinical, cellular and molecular factors. Instances of misuse and over-prescription of antibiotics have become common practices in developing countries [3]. The unlicensed medicine suppliers, uncontrolled antibiotic sales, and availability of over-the-counter antibiotics without a prescription in developing countries, have led to an exponential increase in drug resistance [4]. This dire situation of antimicrobial drug resistance has prompted the search for novel compounds, particularly those from natural sources, with potential antimicrobial properties against various drug-resistant microbial strains.

*Ruta chalepensis* L. (Fam. Rutaceae), commonly known as ‘Fringed rue’, is an Iraqi medicinal plant, endemic to Eurasia and North Africa, and is cultivated elsewhere [5]. This plant is known for its traditional medicinal applications in the treatment of convulsions, dropsy, fever, mental disorders, menstrual problems, microbial infections, neuralgia, rheumatism, and other bleeding and nervous disorders [5,6,7,8]. Previous phytochemical studies on this plant demonstrated the presence of alkaloids, anthraquinones, cardiac glycosides, coumarins, flavonoids, saponins, tannins and terpenoids, whereas pharmacological evaluations established its analgesic, anthelmintic, anti-acetylcholinesterase, anticancer, anti-inflammatory, antimicrobial, antioxidant and antiparasitic properties [5]. We have recently reported the isolation, characterization and antimicrobial activity of three flavonol glycosides, including a new one, 6-hydroxy-rutin 3′,7-dimethyl ether (**16**), from the fruits of this plant [5]. However, to the best of our knowledge, no anti-MRSA (methicillin-resistant *Staphylococcus aureus*) compound was reported from *R. chalepensis* before, and no *in silico* study has been conducted on the antimicrobial compounds from this plant. We now report on the isolation, identification, anti-MRSA activity of several compounds from *R. chalepensis,* collected in Iraq, against a panel of MRSA strains including, ATCC25923 (a standard laboratory strain sensitive to antibiotics like tetracycline), SA1199B, XU212, MRSA-274819 and EMRSA15, and also *in silico* studies on the two most active anti-MRSA compounds from this plant, chalepensin (**10**) and 6-hydroxy-rutin 3′,7-dimethyl ether (**16**).

## 2. Results and Discussion

Leaves, stem, fruits and roots of *R. chalepensis,* collected from Iraq, were Soxhlet extracted separately, but sequentially with *n*-hexane, dichloromethane (DCM) and methanol (MeOH), followed by screening for antimicrobial activity against two Gram-positive (*Staphylococcus aureus* and *Micrococcus luteus*), two Gram-positive bacterial strains (*Escherichia coli* and *Pseudomonas aeruginosa*) and one strain of the fungus *Candida albicans*. All three extracts from leaves, stems and fruits of *R. chalepensis* revealed low to moderate levels of antimicrobial activities against all the tested organisms with varied minimum inhibitory concentration (MIC) values (Table 1). However, in the case of the roots, only the MeOH extract showed activity (MIC 1.25–5 mg/mL) against the test bacteria and fungus (Table 1). Following the initial antimicrobial screening of the crude extracts of *R. chalepensis*, the MICs of 6.25 × 10^−1^ mg/mL was chosen as the minimum threshold of activity for any extract for further analysis, leading to the isolation of compounds responsible for their antimicrobial activity. Many previous studies on *R. chalepensis* documented its antimicrobial activity using different plant parts and methods [9,10,11,12]. However, there is no report on the antimicrobial activity studies of *R. chalepensis* using the modified microtitre assay, or against MRSA strains. Additionally, no *in silico* studies have ever been conducted on anti-MRSA compounds present in this plant.

As the MeOH extracts of different parts of this plant were overall more active than the other solvent extracts, an extensive phytochemical investigation was carried out on the active MeOH extracts of the leaves, stems and fruits of *R. chalepensis* using a combination of reversed-phase solid-phase extraction (SPE) and preparative/semi-preparative HPLC techniques to isolate 18 known secondary metabolites, including alkaloids (**1**–**5**, **12** and **18**), coumarins (**6**–**11**), flavonoid glycosides (**13**–**16**) and a cinnamic acid derivative, 3′′,6′-disinapoylsucrose (**17**), which were identified by HRESIMS and a series of NMR experiments, including ^1^H, ^13^C, DEPTQ, COSY, HSQC and HMBC, as well as by direct comparison with respective spectral data published in the literature. The identified compounds were γ-fagarine (**1**) [13], skimmianine (**2**) [14], kokusaginine (**3**) [15], isokokusaginine (**4**) [16], ribalinium (**5**) [17,18], bergapten (**6**) [19,20], isopimpineline (**7**) [15,21], imperatorin (**8**) [21], chalepin (**9**) [22], chalepensin (**10**) [15], rutamarin (**11**) [15], graveoline (**12**) [23,24,25], rutin (**13**) [5,26,27], rutin 3′-methyl ether (**14**) [5,26,27], rutin 7,4′-dimethyl ether (**15**) [5,28], 6-hydroxy-rutin 3′,7-dimethyl ether (**16**) [5], 3′,6-disinapoylsucrose (**17**) [29,30,31] and arborinine (**18**) [18] (Figure 1). A well-known long-chain alkane (**19**) precipitated out from the *n*-hexane extract of the fruits during extraction.

Of the 19 isolated compounds from *R. chalepensis*, 13 compounds were tested against two Gram-positive (*Staphylococcus aureus* and *Micrococcus luteus*), two Gram-negative (*E. coli* and *Pseudomonas aeruginosa*) bacterial strains and one strain of the fungus *Candida albicans* using the resazurin microdilution method [32] (Table 2). All tested compounds inhibited the growth of *C. albicans* (MIC range: 6.25 × 10^−2^–5 × 10^−1^ mg/mL) and *M. luteus* (MIC range: 6.25 × 10^−2^–5 × 10^−1^ mg/mL), except for compound **19**, which did not show any activity at the highest concentration tested (1 mg/mL) against any of the screened microorganisms. γ-Fagarine (**1**) and bergapten (**6**) failed to prevent the growth of *S. aureus*, but all other tested compounds were active against this bacterial strain with chalepensin (**10**), rutamarin (**11**) and graveoline (**12**) being the most active of the three (MIC: 1.25 × 10^−1^ mg/mL). Whilst compounds chalepin (**9**), chalepensin (**10**), rutin 3′-methyl ether (**14**), 6-hydroxy-rutin 3′,7-dimethyl ether (**16**) and arborinine (**18**) inhibited the growth of *E. coli*, compounds **1**, **3**, **6**–**8** and **11**–**13** failed to inhibit the growth of *E. coli* (Table 2). Compounds **10**, **11**, **13**, **14, 16,** and **18** were found to be moderately active against *P. aeruginosa* with different MIC values (MIC range: 2.5 × 10^−1^–5 × 10^−1^ mg/mL), but all other tested compounds were inactive against this bacterial species. Four compounds including chalepensin (**10**), rutin 3′-methyl ether (**14**), 6-hydroxy-rutin 3′,7-dimethyl ether (**16**) and arborinine (**18**) showed broad-spectrum antimicrobial activity, and were active against all five tested microbial strains (Table 2). The isolation and antimicrobial activity of compounds **14**–**16** against all five microbial strains was reported previously [5].

Thirteen of the 19 isolated compounds from *R. chalepensis* were tested for activity against six MRSA strains (Table 3). The results revealed significant anti-MRSA activity of most of these compounds against the tested strains with different MIC values (64–256 μg/mL). However, γ-fagarine (**1**), bergapten (**6**), isopimpineline (**7**) and graveoline (**12**) did not show any activity against any of the tested MRSA strains (Table 3)**.** Kokusaginine (**3**) and rutin (**13**) were found to be active, albeit at a high concentration (256 μg/mL), only against the MRSA strain MRSA274819, but were inactive against the other five MRSA strains. While chalepin (**9**), chalepensin (**10**) and rutamarin (**11**) are all prenylated furanocoumarin derivatives, they caused different levels of inhibitions because of subtle structural differences. The order of anti-MRSA, the potency of these compounds was **10** > **11** > **9.** Functional groups were the main differences among these three compounds contributing to their differences in lipophilicity. All three compounds are 3-substituted furanocoumarins, among which, except for **10**, the other two compounds are dihydrofuranocoumarins. Rutamarin (**11**), which is simply an acetylated product of chalepin (**9**) was more active than **9**, presumably because of more lipophilicity caused by acetylation. As two other furanocoumarins, bergapten (**6**) and isopimpineline (**7**) were inactive, and none of them has any prenylation at C-3 of the coumarin nucleus like in compounds **9**–**11**, it appears that 3-prenylation is another key determinant of anti-MRSA activity.

6-Hydroxy-rutin 3′-7-dimethyl ether (**16**), rutin (**13**) and rutin 3′-methyl ether (**14**) are flavonoid glycosides that only have differences in the presence/absence and in the number of methyl ether groups in them, offering varying degrees of lipophilicity. Rutin (**13**) does not contain an OMe group, while compounds **14** has an OMe group on 3′ position, and **16** has two OMe groups at positions 3′ and 7. In addition, in **14**, a hydroxyl group occupies position 6. The highest anti-MRSA potency of compound **16** may be because of the different functional groups and their unique positions that make this compound the most lipophilic among these three compounds. The order of anti-MRSA potency in these compounds was **16** > **14** > **13**. This order was also observed in their antimicrobial activity against other test organisms [5]. It is noteworthy that the anti-MRSA activity of compound **16** was quite comparable to that of the positive control norfloxacin (MICs 32 vs. 16 g/mL, respectively). Compound **18** is an acridone alkaloid containing three methyl groups, two of which are oxygenated. This compound has been reported to have many pharmaceutical applications, such as antimicrobial, antiviral, antiplasmodial, antimalarial and anticancer agents, and not surprisingly, as an anti-MRSA agent [33,34,35]. This is the first report on the evaluation of the anti-MRSA effect of isolated compounds, from *R. chalepensis*, against several MRSA strains.

Chalepensin (**10**) and 6-hydroxy-rutin 3′,7-dimethyl ether (**16**), being the most active anti-MRSA compounds found in *R. chalepensis* in the present study, were subjected to *in silico* studies to have an understanding of to what extent these compounds (**10** and **16**) are able to bind to MRSA proteins, and also their drug−like physicochemical characters. The structures of these compounds were optimized (Figure 2) using the Schrodinger suite platform 7.0. [36,37] as the optimization is essential for better understanding interaction patterns and their degree of bonding during the associations. Figure 3, Figure 4, Figure 5 and Figure 6 show the bonded ligands (**10** and **16**) resulting from hydrogen bonding interaction with the target molecules of MRSA, and potential target protein structures of MRSA. To study various bioactivity predictions of the compounds (**10** and **16**), the Circos modelling study was performed with all the functional targeted proteins taken into consideration and the therapeutic potential of the compounds were predicted by the Circos associated prediction model 3.0.1. (Figure 6 and Figure 7) [38]. It can be mentioned here that Circos is a software package for visualizing data and information, and it visualizes data in a circular layout, which makes Circos ideal for exploring relationships between objects or positions. Besides, a circular layout is advantageous, not least being the fact that it is attractive. Associated programming was performed by the Hex server [39]. The results were built up and visualized using the Python server 7.0.1 and the Python with the modeling package. The results of both were screened for the MRSA strains and target proteins like the integrase, penicillin binding proteins (PBPs) [40], pyruvate kinase, and tail−anchored proteins (TaPs) with their specified active site.

In Circos modelling, chalepensin (**10**) demonstrated its predicted bioactivity implicating translational regulation, transportation of the small molecules, membrane proteins and energy metabolism. The important association and activity could deduce its potency as an inflammatory agent (Figure 6). However, the Circos modelling with 6-hydroxy-rutin 3′,7-dimethyl ether (**16**) predicted bioactivity, implicating its chemotaxis role, associated with the DNA replication, modulators of the putative enzymes, membrane proteins and energy metabolism. The important association and activity could deduce its potency as an antireplicative agent and its possible role in the RNA processing and amino-acid biosynthesis. The PASS (Prediction of Activity Apectra for Substances) prediction analysis [41] of compounds **10** and **16** revealed their potency in interacting with various enzymes, associated with various bioactivities as well as potential adverse effects and toxicities (Table 4). It was found that compound **10** could potentially generate itchiness and eye irritation, while compound **16** could trigger metabolic acidosis.

Molecular docking interaction studies on two anti-MRSA compounds, chalepensin (**10**) and 6-hydroxy-rutin 3′,7-dimethyl ether (**16**), against MRSA target proteins revealed their interactions at various levels with integrase, tail−anchored proteins (TaPs), penicillin binding proteins (PBPs) and pyruvate kinase (Table 5), and their hydrogen bonding abilities with those proteins (Table 6). Chalepensin (**10**) was found to possess significant docking ability with PBPs with an e-score of −21.6229 (Figure 8), while that of compound **16** was −9.3219. However, 6-hydroxy-rutin 3′,7-dimethyl ether (**16**) docked better with integrase (−17.331) than compound **10** (−8.933). Both compounds (**10** and **16**) could potentially bind with tail−anchored proteins (TaPs) and pyruvate kinase to similar extents (Table 5).

The active interactive residues include GLN-216 having hydrogen bonding with LYS-218. These amino acid residues mark the integrity of the PBPs and masking of these key residues can offer inactivation of PBPs that are responsible for the ineffectiveness of the strain. It can be noted that the integrase protein is responsible for the efflux of the drug and inactivation of this protein results in inhibition by targeting the residues ASP-111 and ARG-168.

While compound **10** was predicted to be a good blood-brain-barrier (BBB) permeant and CYP2D6 inhibitor, compound **16** was not (Table 7). It can be noted that BBB is one of the parameters that are assessed in *in silico* studies on potential drug molecules to have better understanding of their pharmacology as well as probable toxicity to the brain. However, in the context of anti-MRSA activity, BBB may not be that important, but could be relevant to any probable toxicity of these compounds towards the brain.

Both compounds should have high gastrointestinal (GI) absorption and no violation of the Lipinski rule, which states that an orally active drug has no more than one violation of the following criteria: no more than five hydrogen bond donors (the total number of nitrogen–hydrogen and oxygen–hydrogen bonds), no more than 10 hydrogen bond acceptors (all nitrogen or oxygen atoms), a molecular mass <500 Daltons and an octanol-water partition coefficient that does not exceed five. From the *in silico* studies with anti-MRSA compounds **10** and **16**, it was apparent that these compounds could bind with certain MRSA protein targets, predominantly through hydrogen bonding, as well as van de Waals forces. It was also apparent that these compounds could possess various other bioactivities, as listed in Table 4, with minimum side effects or adverse reactions.

## 3. Materials and Methods

### 3.1. General

Chromatographic solvents were from Fisher Scientific (Loughborough, UK), and used without further purification. The NMR experiments were performed on a Bruker AMX600 NMR spectrometer (600 MHz for ^1^H, and 150 MHz for ^13^C) (BRUKER UK, Coventry, UK). MS analyses were conducted on a Xevo G2-S ASAP (Waters Ltd., Herts, UK) or LTQ Orbitrap XL 1 spectrometers (Thermo Fisher Scientific, Runcorn, UK). UV spectra were obtained on Analytik Jena Specord 210 spectrophotometer (Thermo Fisher Scientific, Runcorn, UK). The solid-phase-extraction (SPE) fractions were analyzed on a Dionex Ultimate 3000 UHPLC, coupled with a photodiode array (PDA) detector, using a Phenomenex Gemini-NX 5 U C_18_ column (150 × 4.6 mm, 5 μm, Phenomenex (Macclesfield, UK), and gradient solvent systems comprising MeOH (solvent B) (Loughborough, UK) and water (solvent A) (both contained 0.1% TFA, flow rate: 1 mL/min) were employed for method developments for preparative HPLC purification. The column temperature was set at 25 °C.

### 3.2. Plant Materials

Leaves, stem bark, fruits and roots of *R. chalepensis* L. were collected from Diyala, Central Iraq (N 33.79684 E 44.623337) during September 2015, air-dried at room temperature, and ground to a fine powder using a coffee grinder. A voucher specimen (No. 33396) for this collection was deposited at the National Herbarium in Iraq.

### 3.3. Extraction

The air-dried ground fruits (103 g), leaves (98 g), stems (81 g) and roots (110 g) of *R. chalepensis* were extracted separately and sequentially with *n*-hexane, dichloromethane (DCM) and methanol (MeOH) (Loughborough, UK) using a Soxhlet apparatus (900 mL, ten cycles each). The crude extracts were concentrated to dryness using a rotary evaporator and stored at 4 °C for further work. Only the MeOH extract showed antimicrobial activity in the initial in vitro antimicrobial screening using resazurin as an indicator of cell growth [5,32], and was subjected to further fractionation, leading to the isolation of antimicrobial compounds.

### 3.4. Initial Antimicrobial Screening

The *n*-hexane, DCM and MeOH extracts of *R. chalepensis* leaves, stems, fruits and roots were initially tested for their antimicrobial activity against two Gram-positive, i.e., *Staphylococcus aureus* (NCTC 12981) and *Micrococcus luteus* (NCTC 7508), two Gram-negative, i.e., *Escherichia coli* (NCTC 12241) and *Pseudomonas aeruginosa* (NCTC 12903), bacterial strains and also a fungal strain, *Candida albicans* (ATCC 90028) using the resazurin 96-well microtitre plate based on in vitro antimicrobial assays [32].

All bacterial and fungal strains were cultured on nutrient agar (Oxoid), followed by incubation at 37 °C for 24 h prior to MIC determination using the resazurin assay. Ciprofloxacin was used as a positive control for bacterial strains, and nystatin for *C. albicans*. Resazurin solution, prepared by dissolving 4 mg of resazurin in 20 mL of sterile distilled water, was used in this assay as an indicator of cell growth. The antimicrobial method used during the study was as described by Reference [32].

Briefly, plates were prepared under aseptic conditions. A sterile 96-well plate was labelled. A volume of 100 µL of test material in 10% (*v*/*v*) DMSO (10 mg/mL for crude extracts) was pipetted into the first row of the plate. To all other wells, 50 µL of normal saline was added. Serial dilutions were performed using a multichannel pipette. Tips were discarded after use, such that each well had 50 µL of the test material in serially descending concentrations. Nutrient broth and 10 µL of resazurin indicator solution was added to each 30 µL well. Finally, 10 µL of bacterial suspension (5 × 10^5^ cfu/mL) was added to each well. Each plate was wrapped loosely with cling film to ensure that bacteria did not become dehydrated. Each plate had a set of controls: a column with a broad-spectrum antibiotic as a positive control (usually ciprofloxacin in serial dilution), a column with all solutions with the exception of the test compound, and a column with all solutions with the exception of the bacterial solution adding 10 µL of nutrient broth instead. The plates were prepared in triplicates and placed in an incubator set at 37 °C for 18–24 h. The color change was then assessed visually. Any color changes from purple to pink was recorded as positive. The lowest concentration at which color change occurred was taken as the MIC value. The average of three values was calculated and that was the MIC for the test material and bacterial strain.

### 3.5. Solid-Phase Extraction (SPE)

A portion (2 g) of the active MeOH extract of each plant part was subjected to SPE on a Strata C_18_ reversed-phase cartridge (20 g, Phenomenex, Macclesfield, UK), eluted with a step-gradient using water-MeOH mixture of decreasing polarity, water:MeOH 80:20, 50:50, 20:80 and 0:100 (200 mL each), to obtain four SPE fractions I-IV, respectively. All SPE fractions were dried using a rotary evaporator followed by freeze-drying and stored in sealed vials in a fridge at 4 °C for further work.

### 3.6. Isolation and Identification of Compounds

Both of fractions II (50% MeOH in water) and III (80% MeOH in water) of leaves and fruits were subjected to preparative HPLC using an ACE prep-column (150 × 21.2 mm, 5 μm, Hichrom Ltd., UK; MeOH-water linear gradient, flow rate: 10 mL/min, monitored simultaneously at 215, 254, 280 and 320 nm), whilst a semi-prep Phenomenex Gemini-NX 5 U C_18_ column (150 × 4.6 mm, 5 μm, Phenomenex (Macclesfield, UK); MeOH-water linear gradient, flow rate: 2 mL/min, monitored simultaneously at 215, 254, 280 and 320 nm) have been used to separate compounds from stem fractions II and III. An Agilent 1260 Infinity series preparative HPLC, coupled with a PDA detector, was used to isolate compounds. The mobile phase comprised solvents A (0.1% TFA in HPLC grade water) and B (0.1% TFA in HPLC grade methanol) operated on gradient system (30–100% A in B for 30 min followed by 100% A for 10 min and finally 100–30% A in B for 5 min). Preparative HPLC analysis of the SPE fraction II of the fruit extract using the above method yielded compounds **5** (0.5 mg; *t*_R_ 8.3 min), **13** (1.2 mg; *t*_R_ 16.02 min), **14** (1.0 mg; *t*_R_ 16.99 min), **16** (0.3 mg; *t*_R_ 17.78 min), **12** (1.2 mg; *t*_R_ 18.68 min), **7** (1.0 mg; *t*_R_ 21.43 min), **1** (0.4 mg; *t*_R_ 22.73 min), **2** (3 mg; *t*_R_ 22.4 min) and **3** (2 mg; *t*_R_ 22.5 min). Whereas, the identical preparative HPLC on SPE fraction III of the fruit extract gave more of compounds **13** (0.6 mg) and **2** (0.5), in addition to **8** (1.8 mg; *t*_R_ 25.15 min), **10** (0.2 mg; *t*_R_ 26.37 min), **9** (1.2 mg; *t*_R_ 26.92 min) and **11** (1.5 mg; *t*_R_ 27.94 min). Similarly, preparative HPLC using the above method on the SPE fraction II of the leaves methanolic extract produced a total of eight compounds including **6** (1.3 mg; *t*_R_ 13.34 min), **13** (0.3 mg), **1** (0.3 mg), **3** (1.0 mg), **14** (0.5 mg), **15** (0.3 mg; *t*_R_ T 17.03 min), **4** (1.5 mg; *t*_R_ 19.3 min) and **12** (1.2 mg). Furthermore, the separation process of fraction III of the leaves methanolic extract afforded more of compounds **13** (0.2 mg), **12** (1.3 mg), **6** (0.2 mg), **3** (1.5 mg), **1 (**0.5 mg), **9** (1.2 mg) and **10** (0.2 mg), in addition to **18** (0.3 mg; *t*_R_ 20.05 min). Moreover, the semi-preparative HPLC analysis of the SPE fractions II and III on stem methanolic extract, using the same gradient of 30–100% MeOH in water (2 mL/min) over 30 min, led to the isolation of six compounds including **17** (1.0 mg; *t*_R_ 11.56 min), and more of **13** (0.8 mg; *t*_R_ 12.5 min), **15** (0.2 mg; *t*_R_ 14.7 min), **3** (0.9 mg; *t*_R_ 20.64 min), **1** (0.4 mg; *t*_R_ 21.11 min), **6** (1.6 mg; *t*_R_ 22.38 min), **18** (0.7 mg; *t*_R_ 23.01 min), **9 (**0.2 mg; *t*_R_ 27.03 min), and **10** (0.4 mg; *t*_R_ 28.5 min). Compound **19** precipitated from the fruit *n*-hexane extract during extraction. The chemical structures of these compounds were confirmed by 1D (^1^H, ^13^C, DEPTQ) and 2D (COSY, NOESY, HSQC and HMBC) NMR spectroscopy and mass spectrometry, and by comparison with respective published data.

### 3.7. Resazurin Assay with Isolated Compounds

The modified resazurin assay, as described by Sarker et al. [32], was used to determine antimicrobial activity and the MIC values, and where appropriate of 14 isolated compounds (**1**, **3**, **6**–**14**, **16**, **18** and **19**) against *Escherichia coli* (NCTC 12241), *Micrococcus luteus* (NCTC 7508), *Pseudomonas aeruginosa* (NCTC 12903), *Staphylococcus aureus* (NCTC 12981) and *Candida albicans* (ATCC 90028) [5]. Four other compounds (**2**, **4**, **5** and **17**) were not tested because of the paucity of samples.

### 3.8. Assessment of Anti-MRSA Activity

All chemicals for the anti-MRSA assay was purchased from Sigma-Aldrich (Gillingham, UK), Cation-adjusted Mueller-Hinton broth was sourced from Oxoid Microbiology Products, UK, and was adjusted to have 20 and 10 mg/L of Ca^2+^ and Mg^2+^ ions, respectively. The *Staphylococcus aureus* strains used in this study were ATCC25923 (a standard laboratory strain sensitive to antibiotics like tetracycline), SA1199B, XU212, MRSA-274819, MRSA340702 and EMRSA15 [9]. SA1199B overexpresses the NorA MDR efflux pump [42] and XU212 is a Kuwaiti hospital isolate that is a MRSA strain possessing the TetK tetracycline efflux pump [9], whereas the EMRSA 15 strain [43] was epidemic in the UK. All these were obtained from the National Collection of Type Cultures (NCTC). The assay protocol was exactly as described by Nurunnabi et al. [44]. Norfloxacin, a well-known antibiotic, was used as the positive control.

Briefly, an inoculum density of 5 × 10^5^ colony-forming units of each bacterial strain was prepared in normal saline (9 g/L) by comparison to a 0.5 MacFarland turbidity standard. The inoculum (125 μL) was added to all wells, and the microtitre plate was incubated at 37 °C for the corresponding incubation time. For MIC determination, 20 μL of a 5 mg/mL methanolic solution of 3-[4,5-dimethylthiazol-2-yl]-2,5-diphenyltetrazolium bromide (MTT) was added to each of the wells and incubated for 20 min. Bacterial growth was indicated by a color change. The minimum inhibitory concentrations (MICs) were determined using the broth microdilution method according to National Committee for Clinical Laboratory Standards with modification using nutrient broth as the medium.

### 3.9. In Silico Studies with Two Most Active Anti-MRSA Compounds from This Plant, Chalepensin *(**10**)* and 6-Hydroxy-rutin 3′,7-dimethyl Ether *(**16**)*

*In silico* studies with two most active anti-MRSA compounds from this plant, chalepensin (**10**) and 6-hydroxy-rutin 3′,7-dimethyl ether (**16**), were conducted using a variety of methods and protocols, as described in the Results and Discussion Sections earlier. Briefly, the structures of these compounds were optimized using the Schrodinger suite platform 7.0. (Schrodinger, Cambridge, UK) [36,37]. To study the various bioactivity prediction of the compounds (**10** and **16**), the Circos modelling study was performed with all the targeted proteins taken into consideration and the therapeutic potential of the compound were predicted by the Circos associated prediction modelling 3.0.1. (Echelon Innovation Centre, Vancouver, Canada) [38]. Associated programming was carried out with the Hex server [39]. The results were built up and visualized using the Python server 7.0.1 and the Python with the modeling package. The PASS prediction [41] analysis was used to predict the potency of these compounds in interacting with various enzymes, associated with various bioactivities as well as potential adverse effects and toxicities.

## 4. Conclusions

The present work generated the first comprehensive phytochemical report on the analysis of Iraqi *R. chalepensis* species, along with their antimicrobial activity using the modified microtitre assay. This is also the first report on the antibacterial activity of the compounds isolated from *R. chalepensis* against a panel of methicillin-resistant *Staphylococcus aureus* (MRSA). The outcome of this study demonstrated that at least seven of the isolated compounds from various parts of *R. chalepensis* possess reasonable anti-MRSA property. Among the active compounds, chalepensin (**10**), and 6-hydroxy-rutin 3′,7-dimethyl ether (1**6**) appear to be the most active compounds from *R. chalepensis* and are active against four out of the six tested MRSA strains. *In silico* studies on compounds **10** and **16** revealed that both compounds should have high GI absorption and no violation of the Lipinski rules, meaning ‘drug-like’ characters in these compounds, and it was also apparent that these compounds could bind with certain MRSA protein targets, predominantly through hydrogen bonding as well as van de Waals forces. Based on the current findings, it can be assumed that these two compounds might be utilized as structural templates for generating structural analogues and developing potential anti-MRSA therapeutic agents.

## Figures and Tables

**Figure 1 molecules-26-01114-f001:**
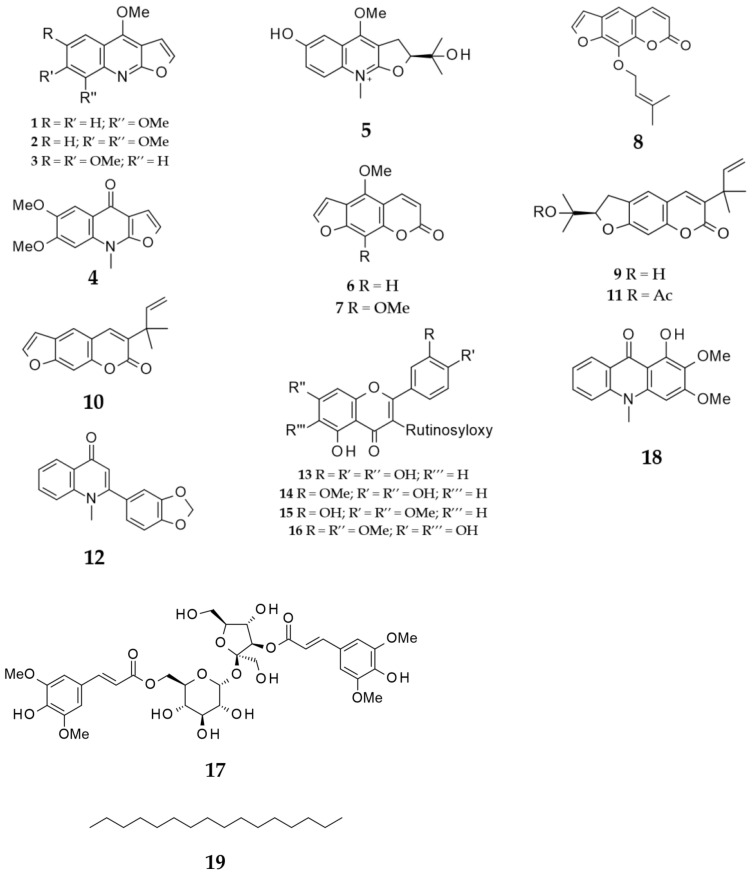
Isolated compounds from the Iraqi *R. chalepensis.*

**Figure 2 molecules-26-01114-f002:**
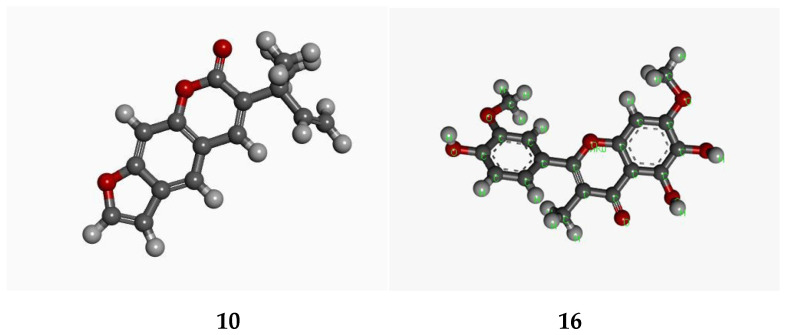
Optimized structures of chalepensin (**10**) and 6-hydroxy-rutin 3′,7-dimethyl ether (**16**).

**Figure 3 molecules-26-01114-f003:**
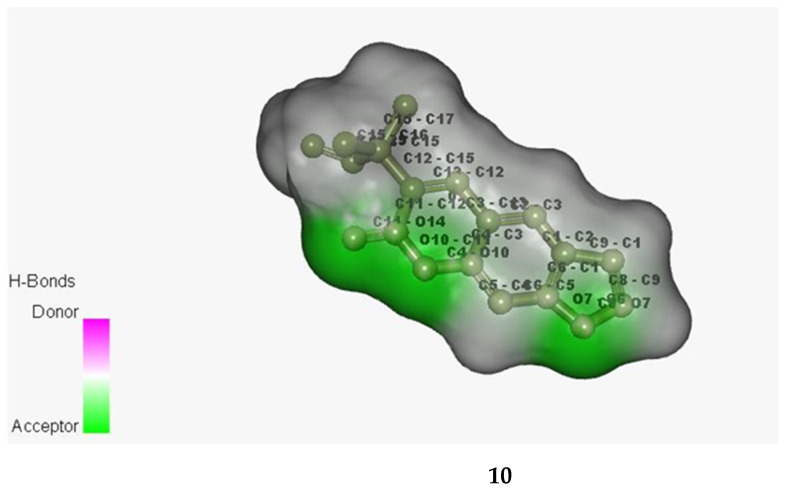
Bonded ligands (**10** and **16**) resulting from hydrogen bonding interaction with the target molecule of MRSA.

**Figure 4 molecules-26-01114-f004:**
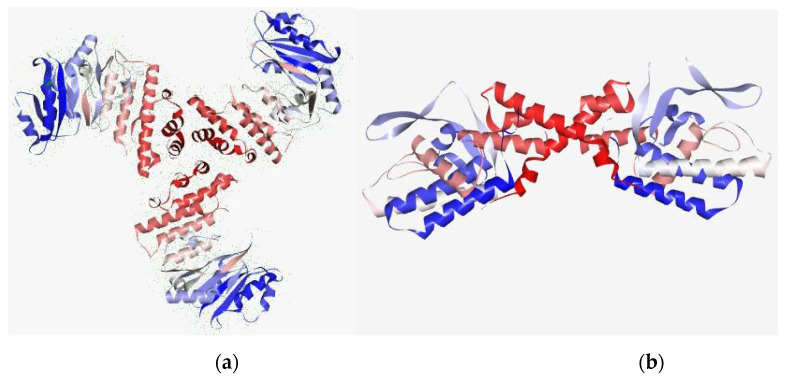
Potential target of the MRSA (target protein): (**a**) secondary structure from N to C terminal of TaP and (**b**) integrase. Blue color indicates N terminal and red color indicates C terminal. The protein is a depiction of the 3D modelled structure indicating the secondary structures in shades.

**Figure 5 molecules-26-01114-f005:**
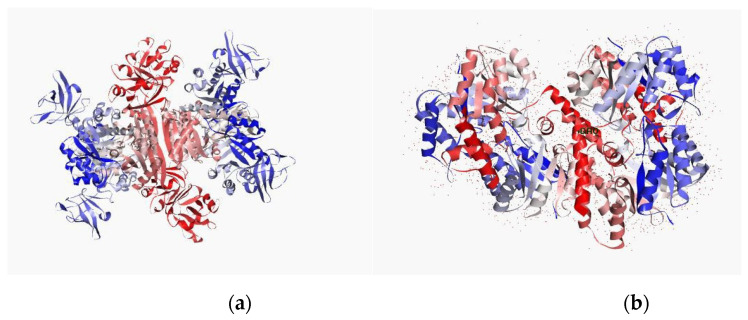
Potential target of the MRSA (target protein): (**a**) secondary structure from N to C terminal of pyruvate kinase and (**b**) penicillin-binding protein. Blue color indicates N terminal and red color indicates C terminal. The protein is a depiction of the 3D modelled structure indicating the secondary structures in shades.

**Figure 6 molecules-26-01114-f006:**
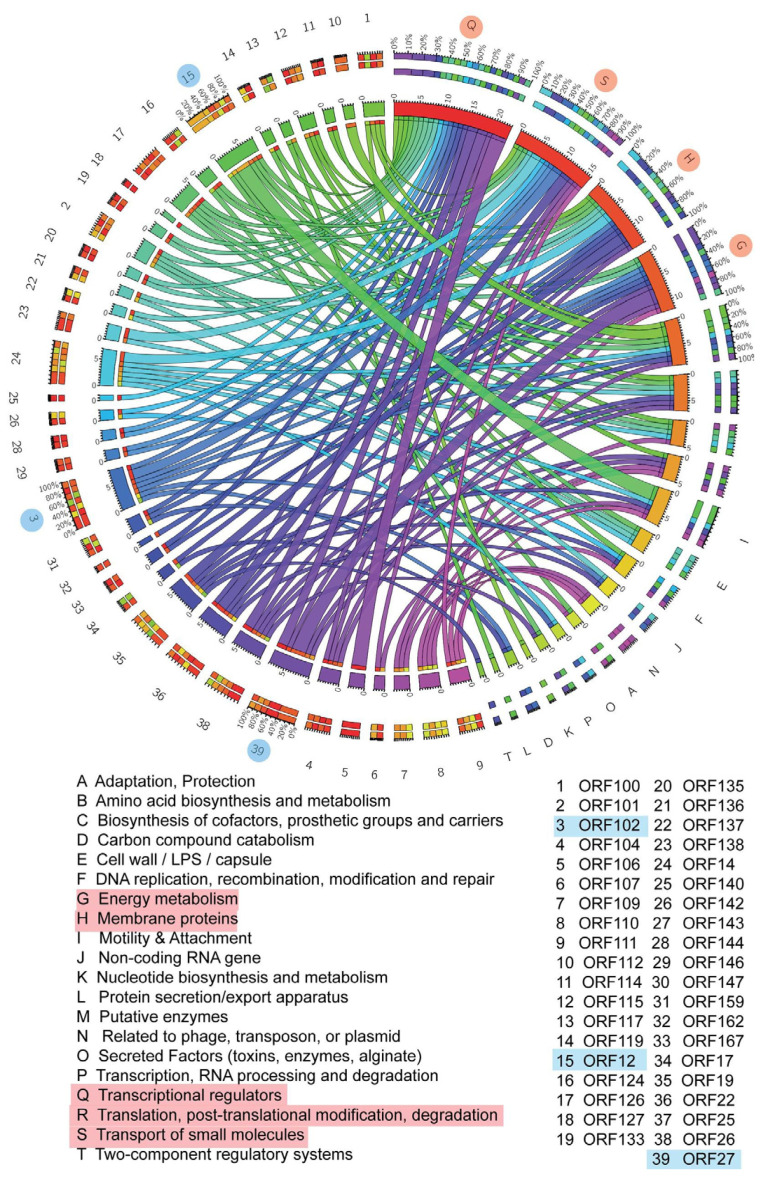
Circos modelling of chalepensin (**10**) with its predicted bioactivity implicating its translational regulators, transportation of the small molecules, membrane proteins, energy metabolism. The important association and the activity deduce its potency as inflammatory activity. The highlighted ORF and its respective functions indicate cellular processes that are positively correlated with physiological processes contributing to the anti-MRSA activity. Different colors indicate various pivotal cellular processes shown by different letters and different gradients explain the degree of the interrelated network.

**Figure 7 molecules-26-01114-f007:**
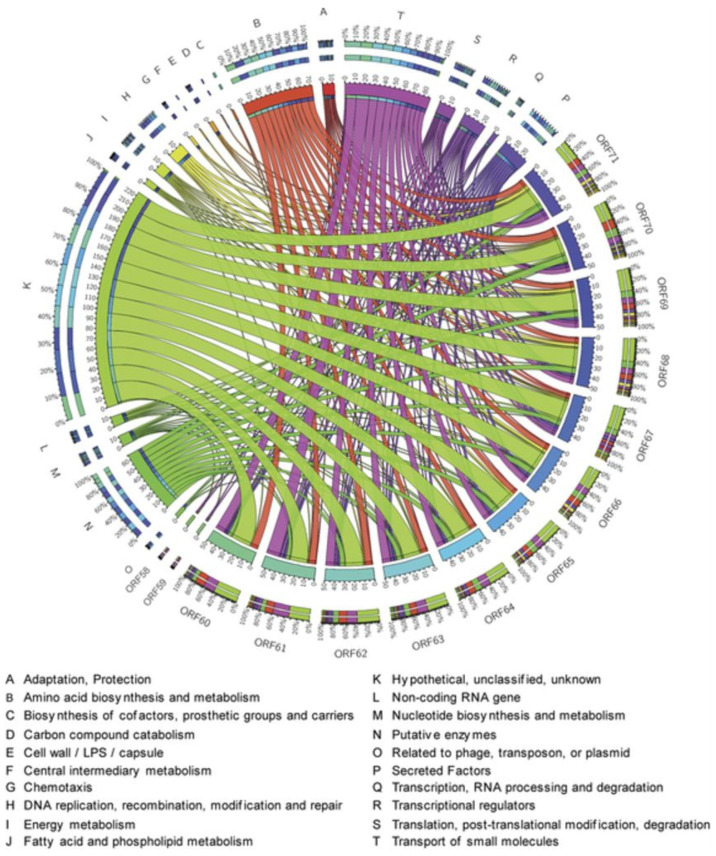
Circos modelling of 6-hydroxy-rutin 3′,7-dimethyl ether (**16**) with its predicted bioactivity implicating chemotaxis role, associated with the DNA replication, modulators of the putative enzymes, membrane proteins and energy metabolism. The important association and the activity deduce its potency as antireplicative property and its role in the RNA processing and amino-acid biosynthesis. Different colors indicate various pivotal cellular processes shown by different letters and different gradients explain the degree of the interrelated network.

**Figure 8 molecules-26-01114-f008:**
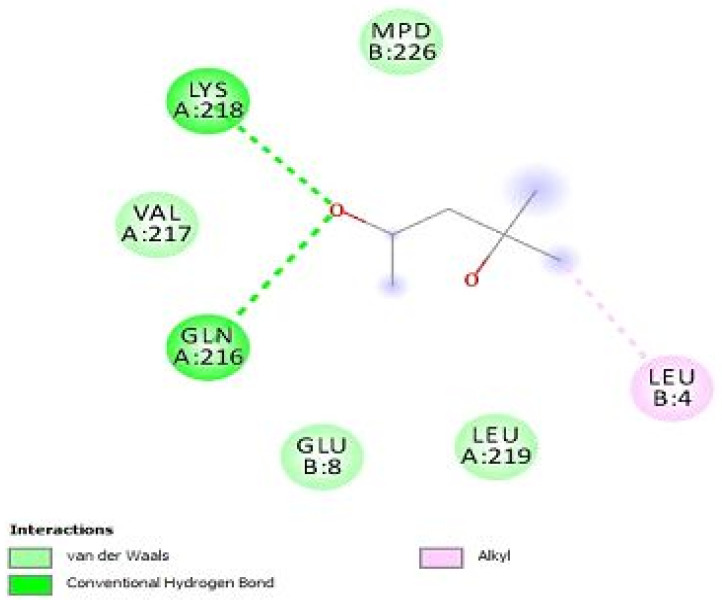
Dock poses of chalepensin (**10**) with the penicillin-binding protein (PBPs) with an e-score of −21.6229. The interactive residues with an active pocket show the presence of all three pivotal bonds with Van der Waals interaction, hydrogen bonding and the alkyl bonding.

**Table 1 molecules-26-01114-t001:** Antimicrobial activity of the *n*-hexane, DCM and MeOH extracts of various plant parts (leaves, stem, fruits and roots) of *Ruta chalepensis *.*

Bacteria and Fungi		Extract	Plant Parts (MICs in mg/mL)			
			Leaves	Stem	Fruits	Roots
**Gram-Negative** **Bacteria**	*E. coli*	*n*−Hexane	6.25 × 10^−1^	6.25 × 10^−1^	3.12 × 10^−1^	N/A
		DCM	3.12 × 10^−1^	6.25 × 10^−1^	3.12 × 10^−1^	N/A
		Methanol	6.25 × 10^−1^	3.12 × 10^−1^	6.25 × 10^−1^	5
	*P. aeruginosa*	*n*−Hexane	6.25 × 10^−1^	6.25 × 10^−1^	3.12 × 10^−1^	N/A
		DCM	3.12 × 10^−1^	6.25 × 10^−1^	6.25 × 10^−1^	N/A
		Methanol	6.25 × 10^−1^	3.12 × 10^−1^	1.56 × 10^−1^	2.5
**Gram-Positive** **Bacteria**	*M. luteus*	*n*−Hexane	1.56 × 10^−1^	6.25 × 10^−1^	3.12 × 10^−1^	N/A
		DCM	1.56 × 10^−1^	6.25 × 10^−1^	7.81 × 10^−2^	N/A
		Methanol	1.95 × 10^−2^	7.81 × 10^−2^	3.90 × 10^−2^	1.25
	*S. aureus*	*n*−Hexane	6.25 × 10^−1^	6.25 × 10^−1^	6.25 × 10^−1^	N/A
		DCM	6.25 × 10^−1^	6.25 × 10^−1^	6.25 × 10^−1^	N/A
		Methanol	3.12 × 10^−1^	3.12 × 10^−1^	3.12 × 10^−1^	5
**Pathogenic Fungi**	*C. albicans*	*n*−Hexane	6.25 × 10^−1^	1.95 × 10^−1^	1.56 × 10^−2^	N/A
		DCM	3.12 × 10^−1^	3.12 × 10^−1^	3.90 × 10^−2^	N/A
		Methanol	1.95 × 10^−2^	7.81 × 10^−2^	3.90 × 10^−2^	2.5

* Positive controls used were ciprofloxacin against bacterial strains and nystatin against the fungal strain, and the MIC values are shown in Table 2.

**Table 2 molecules-26-01114-t002:** Antibacterial activity of compounds isolated from *Ruta chalepensis.*

Compounds	MIC in mg/mL				
	*Staphylococcus aureus* NCTC 12981	*Escherichia coli* NCTC 12241	*Pseudomonas aeruginosa* NCTC 12903	*Micrococcus luteus* NCTC 7508	*Candida albicans* ATCC 90028
**1**	N/A	N/A	N/A	5 × 10^−1^	5 × 10^−1^
**3**	5 × 10^−1^	N/A	N/A	5 × 10^−1^	5 × 10^−1^
**6**	N/A	N/A	N/A	5 × 10^−1^	5 × 10^−1^
**7**	5 × 10^−1^	N/A	N/A	6.25 × 10−2	5 × 10^−1^
**8**	5 × 10^−1^	N/A	N/A	2.5 × 10^−1^	2.5 × 10^−1^
**9**	1.25 × 10^−1^	5 × 10^−1^	N/A	1.25 × 10^−1^	1.25 × 10^−1^
**10**	1.25 × 10^−1^	1.25 × 10^−1^	5 × 10^−1^	2.5 × 10^−1^	6.25 × 10^−2^
**11**	1.25 × 10^−1^	N/A	5 × 10^−1^	2.5 × 10^−1^	2.5 × 10^−1^
**12**	5 × 10^−1^	N/A	N/A	5 × 10^−1^	5 × 10^−1^
**13**	2.5 × 10^−1^	N/A	5 × 10^−1^	2.5 × 10^−1^	2.5 × 10^−1^
**14**	2.5 × 10^−1^	2.5 × 10^−1^	2.5 × 10^−1^	2.5 × 10^−1^	2.5 × 10^−1^
**16**	2.5 × 10^−1^	2.5 × 10^−1^	2.5 × 10^−1^	6.25 × 10^−2^	6.25 × 10^−2^
**18**	2.5 × 10^−1^	5 × 10^−1^	5 × 10^−1^	1.25 × 10^−1^	6.25 × 10^−2^
Ciprofloxacin	9.76 × 10^−4^	1.55 × 10^−2^	1.95 × 10^−3^	9.76 × 10^−4^	N/A
Nystatin	N/A	N/A	N/A	N/A	9.76 × 10^−4^

**Table 3 molecules-26-01114-t003:** Anti-MRSA activity of isolated compounds from *Ruta chalepensis* against clinical isolates of methicillin-resistant *Staphylococcus aureus.*

Compounds			MIC in μg/mL			
	XU212	ATCC25923	SA1199B	EMRSA-15	MRSA346702	MRSA274819
**1**	−	−	−	−	−	−
**3**	−	−	−	−	−	256
**6**	−	−	−	−	−	−
**7**	−	−	−	−	−	−
**8**	256	−	256	−	256	256
**9**	−	256	256	−	−	128
**10**	64	128	−	−	64	64
**11**	128	−	128	−	128	128
**12**	−	−	−	−	−	−
**13**	−	−	−	−	−	256
**14**	256	128	−	−	256	256
**16**	32	64	−	−	128	256
**18**	−	256	−	128	64	256
Norfloxacin	16	2	32	1	64	64

**Table 4 molecules-26-01114-t004:** PASS prediction (bioactivity and toxicity) analysis of chalepensin (**10**) and 6-hydroxy-rutin 3′,7-dimethyl ether (**16**).

Bioactivities	Pa (Probability to be Active)		Pi (Probability to be Inactive)	
	10	16	10	16
Membrane integrity agonist	0.954	0.973	0.003	0.002
Ubiquinol-cytochrome-c reductase inhibitor	0.863	−	0.013	−
Fatty-acyl-CoA synthase inhibitor	0.822	−	0.004	−
Membrane permeability inhibitor	0.809	0.962	0.009	0.002
Free radical scavenger/antioxidant	0.627	0.878	0.005	0.003
Alcohol dehydrogenase (NADP^+^) inhibitor	−	0.927	−	0.002
Xenobiotic-transporting ATPase inhibitor	−	0.886	−	0.002
Lipid peroxidase inhibitor	−	0.813	−	0.003
Anticarcinogenic	−	0.716	−	0.007
Toxicity/adverse reactions	Compound **10**: itchiness and eye irritation acidosis		Compound **16**: metabolic acidosis	

**Table 5 molecules-26-01114-t005:** Molecular docking interactions of chalepensin (**10**) and 6-hydroxy-rutin 3′,7-dimethyl ether (**16**) with MRSA protein targets and the docking scores.

Anti-MRSA Compounds	Integrase	Tail-Anchored Proteins (TaPs)	Penicillin Binding proteins (PBPs)	Pyruvate Kinase
**10**	−8.933	−11.8567	−21.6229	−13.615
**16**	−17.331	−11.8148	−9.3219	−16.237

**Table 6 molecules-26-01114-t006:** Hydrogen bonding properties of chalepensin (**10**) and 6-hydroxy-rutin 3′,7-dimethyl ether (**16**) with MRSA protein targets.

anti-MRSA Compounds	Score	Hydrogen Bonding Properties		
		Bond Attributes	Bond Energy	Bond Length (Å)
**Integrase**				
**10**	−8.933	O39-HH11-ARG-68-B H78-O-H15-108-B O53-HE21-GCN-109-B	−0.8	1.80
			−3.3	2.09
			−2.6	4.22
**16**	−17.331	H84-O-ASP-107-B H79-OD1-ASP-111-B O6-HH21-ARG-16-8	−4.7	2.20
			−3.7	2.23
			−4.2	1.77
**TaPs**				
**10**	−11.8567	H39-OE1-GLN-109-B O23-HE21-GLN-109-B	−4.1	1.99
			−4.1	2.01
**16**	11.8148	O17-HH21-ARG-68-B O16-HE21-GLN-53-B O20-HH11-ARG-68-B O20-HH21-ARG-68-B	−6.3	2.02
			−2.3	2.32
			−3.7	2.00
			−3.5	2.06
**PBPs**				
**10**	−21.6229	O1-HH21-LYS-218-B O14-H-GLN-216-B	−7.3	2.13
			−7.6	1.70
**16**	−9.3219	O14-H-GLN-216-B H21-002-ASP-111- H21-001-ASP-111-B	−3.0	1.96
			−4.6	2.14
			−4.2	2.18
**Pyruvate Kinase**				
**10**	−13.615	O16-HH21-ARG-67-B	−2.1	2.37
**16**	−16.237	O3-HE22-GLN-53-B O16-HH11-ARG-16-B O16-HH21-ARG-16-B O19-H-ALA-13-B	−4.7	1.96
			−5.5	2.15
			−6.9	2.12
			−4.2	2.13

**Table 7 molecules-26-01114-t007:** Interaction of chalepensin (**10**) and 6-hydroxy-rutin 3′,7-dimethyl ether (**16**) with CYPs, solubility and gastrointestinal (GI) absorption, as determined by the Silicos-IT chemoinformatic software (silicos-it.be.s3-website-eu-west-1.amazonaws.com, accessed on 7 February 2021).

Compds	Silicos-IT Log Sw	Silicos-IT Solubility		Silicos IT Class	GI Absorption	BBB Permeant	PGP Substrate	CYPs Inhibitor					Log Kp (cm/s)	Lipinski Violations
		mg/mL	Mol/L					1A2	2C19	2C9	2D6	A4		
**10**	−2.93	3.41 × 10^−1^	1.18 × 10^−3^	Soluble	High	Yes	No	No	No	No	Yes	No	−6.77	0
**16**	−1.70	3.28	2.00 × 10^−2^	Soluble	High	No	No	No	No	No	No	No	−7.29	0

## Data Availability

All relevant data have been presented as an integral part of this manuscript.

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
