# Peer review of "Anti-MRSA Constituents from Ruta chalepensis (Rutaceae) Grown in Iraq, and In Silico Studies on Two of Most Active Compounds, Chalepensin and 6-Hydroxy-rutin 3′,7-Dimethyl ether"

_molecules, 2021, doi:10.3390/molecules26041114_

Round 1

Reviewer 1 Report

In this manuscript, the have isolated several compounds from Ruta chalepensis (Rutaceae) and characterised them using a combination of spectroscopic techniques and in comparison with literature data for the known compounds. The compounds were, in turn, evaluated for biological activity in vitro against bacterial (Gram positive and negative) and fungal strains. Molecular docking (in silico) studies were also performed on two of the most active compounds from the series, namely, chalepensin and 6-hydroxy-rutin 3',7-dimethyl ether. The authors have also discussed some structure activity properties for some of the related compounds. The article is well written and relatively free of language or typographical errors. The results of this study worth publication in Molecules pending some corrections as indicated below.

Title:

This is succinct and in line with the context of this investigation.

Abstract:

No errors detected and presents the context of this study well. Perhaps the authors should write MRSA in full here as this acronym is used again in the Keywords without definition.

Introduction

The authors have provided literature background with relevant citations to lead to the problem statement or rationale for this study. However, I have detected a repetition between the sentence that starts from lines 63-65, and those in lines 83-87. I could not understand why the authors used citation [9] to the corresponding sentence as if they performed this study before. The study in this cited paper was performed by another group many years back and not on this plant species. The cited paper is relevant elsewhere in the manuscript.

I suggest that the authors move sentences in lines 83-87 to replace the entire sentence that starts from lines 63-65: ‘However, to the best of our knowledge…’ Reference [9] should be cited where relevant.

Results and discussion

The authors should check spacings between substituent numbers for the names of compounds in this section and Experimental section. Also attend to spacings between reference numbers in the text for consistency.

Lines 118-121: Rephrase the sentence to exclude ‘Among the tested compounds from R. Chalepensis, four compounds including’ to start with compound names.

Line 131-133: Since two compounds, ‘but were inactive against other five MRSA strains.’

Lines 162-167: The authors talk about the many biological applications of compound 18 and that its anti-MRSA activity has not been evaluated before. Subsequent sentence talks about all the tested compounds in general. The reader has to refer many pages back to see that 18 has also been evaluated for activity. I suggest that the authors combine the two sentences on lines 162-165 into one to describe the compound and its application ending with citations [33-35]. THn the authors should have a separate sentence to indicate that compound 18 has only been evaluated for anti-MRSA for the 1st time. The sentence starting from line 165-167 comes a few times in the Abstract and Introduction as well as Results and Discussion. Tis sentence could fit well under Conclusions.

The authors should elaborate and justify the need for performing molecular docking separately and then mention the software they used. The sentence in its current form is too vague as it refers to optimization of the structures and not ligand-receptor interactions.

Line 234. The authors should define the acronym ‘BBB’, I guess ‘blood brain barrier’  and try to justify the relevance of permeability of the anti-MRSA through this membrane in particular.

Materials and methods and Conclusions

I am happy with the way this part is presented

References

The authors should be consistent in the presentation of journal names. In some cases they use symbol for ‘and’ or in full. The use of journal name abbreviations would be more appropriate.

With all the suggested changes implemented, I would recommend acceptance and publication of this manuscript in Molecules.

Reviewer 2 Report

Authors have experimented and submitted a research article entitled Anti-MRSA constituents from Ruta chalepensis (Rutaceae) grown in Iraq, and in silico studies on two most active compounds, chalepensin and 6-hydroxy-rutin 3', 7-dimethyl ether. The experimental protocol and experimentation are scientifically well. Data presentation and its interpretation are also acceptable. It is essential to identify and develop natural active ingredients against antibiotic-resistant bacteria. 

Authors have initially analyzed the antibacterial activity of different extracts such as hexane, DCM, and MeOH. This data shows all the extract showed good antibacterial activity in particular DCM extract showed more potent than the others, but authors selected MeOH for in-depth analysis. Please explain in details.

Should be maintained uniform units for concentrations of compounds used in antibacterial activity, some places authors mentioned as X^ mg/mL or ug/mL for MIC experiment.

Provide detailed legends for all the tables and figures 

Should provide a clear interpretation of results with easily understandable (γ-Fagarine (1) and bergapten (6) failed to prevent the growth of S. aureus, but all other tested compounds), simply interpret as Fagarine (1) and bergapten (6) did not exhibit or fail to prevent the growth of all tested bacteria etc.…

Reviewer 3 Report

Should be published in Molecules after major revision

In their manuscript entitled “Anti-MRSA constituents from Ruta chalepensis (Rutaceae) grown in Iraq, and in silico studies on two most active compounds, chalepensin and 6-hydroxy-rutin 3',7-dimethyl ether”, Shaymaa Al-Majmaie et al. report the evaluation of a series of compounds, including their activity against a panel of bacteria (especially MRSA strains), and in silico analyses to get insights into the properties of most active ones.

Overall appreciation

In my opinion, this paper provides various interesting results but, unfortunately, it is not easy to read and understand. The methods would need to be more adequately described, especially those related to in silico studies. Unnecessary duplications should be avoided whereas additional explanations are needed, in the text as well as in the captions of Figures and Tables. Authors emphasized throughout their manuscript on a so-called anti-MRSA activity of the bioactive compounds isolated. This can suggest some greater activity against such bacteria, which is not really demonstrated. Finally, authors should not overstate their findings, notably in Conclusion lines 394-395: “The outcome of this study demonstrated that at least seven of the isolated compounds from various parts of R. chalepensis possess considerable anti-MRSA property”. In summary, this paper needs a thorough revision for addressing various issues, which could noticeably enhance its quality and increase its impact. For all these reasons, I would recommend publication of this manuscript in Molecules after a major revision have been done, by taking into account the above-mentioned comments and those detailed below.

Detailed comments

Extraction, isolation and identification of compounds: Detailed spectroscopic data of the isolated compounds are not provided (they could be given as supplementary material). By the way, it is unclear to me whether these compounds were partly or completely reported before, especially in [5]. Please clarify and adapt the manuscript accordingly.

Antimicrobial assays: The range of concentrations tested for determining MIC is not clearly mentioned. In Table 1, some controls are missing, especially the effect of solvents when used alone. The strains evaluated in Table 1 must be specified (not only mentioning microbial species). MIC values in Tables 1, 2 and 3 should be provided using the same units (in µg/mL or in µM, as typically used). There is no information about technical and biological replicates. Bacteriostatic and/or bactericidal effects should have been determined to better characterize the antimicrobial effects reported.  

In silico studies: Many explanations are missing. It looks like this paper was written assuming that readers are familiar with the various bio-informatics analyses conducted, which is not obvious. In Materials and Methods, the corresponding section is the shortest. In Table 4, what does mean Pa? Pi? In Figures 6 and 7, it is unclear why some items are highlighted (e.g. ORF102, 12 and 35). What is the color code? GI absorption, Lipinski rules and other terms notably used in Table 7 are not defined. Data in Figures 6, 7, 8 and Tables 4, 5, 6 and 7 look only partly exploited. Some of them could be moved to supporting information online, where more explanations would be provided.

The MSSA laboratory strain RN4220 is erroneously mentioned once in the Abstract line 27 as a MRSA strain. It is not mentioned thereafter in the manuscript, which suggests it is an error. Please revise.

Line 103, Authors state: “Of the 19 isolated compounds from R. chalepensis, 15 compounds were tested […]”, referring then to Table 2. However, this Table summarizes the results of 13 (not 15) compounds from R. chalepensis.

Some typo errors must be corrected. For instances, line 114: “E. Coil”; line 115: “E. Coli”.

Please define all abbreviations used.

Round 2

Reviewer 3 Report

Response to revision:

Responses are in bold and underlined.

“In my opinion, this paper provides various interesting results but, unfortunately, it is not easy to read and understand.”

Response: The reviewer is fully entitled to his/her opinion, right or wrong! However, I am afraid, we strongly disagree with this statement. If he/she fails to understand any particular section, I am happy to explain it to him/her. Note that none of the other reviewers found the paper difficult to understand!! We believe that the manuscript is easy to understand in its present form.

As a reviewer, I feel entitled to point out to authors that my opinion may be shared by other readers. For this reason, authors should consider it with attention, independently of the other reviews done. I (as well as the other reviewers) spent some time to review this paper, with the aim to help improve its quality. Thus I would have appreciated receiving respectful answers and acknowledgment.

“The methods would need to be more adequately described, especially those related to in silico studies.”

Response: All methods applied in this paper have been described adequately with appropriate references. As the in silico methods are described in the Results and Discussion section as part of the discussion with all required references, we do not believe that any duplication is necessary in the Experimental section.

“Unnecessary duplications should be avoided whereas additional explanations are needed, in the text as well as in the captions of Figures and Tables.”

Response: Unless specified, we simply cannot find any ‘unnecessary duplication’ in anywhere in our paper. In fact, we consciously tried to avoid duplication, Hence, we did not duplicate in silico method in the Experimental. If there is any minor duplication anywhere, that is for a reason, definitely not ‘unnecessary’.

For instance: the sentence in lines 63-65 (“However, to the best of our knowledge, no anti-MRSA compound was reported from R. chalepensis before, and no in silico study has been conducted on the antimicrobial compounds from this plant”) is almost the same as the sentences in lines 84-87 (“However, there is no report on the antimicrobial activity studies of R. chalepensis using the modified microtitre assay, or against MRSA strains. Also, no in silico studies have ever been conducted on anti-MRSA compounds present in this plant.”).

“Authors emphasized throughout their manuscript on a so-called anti-MRSA activity of the bioactive compounds isolated.”

Response: The paper is about anti-MRSA activity (not so-called!), and naturally that should be emphasized. What is wrong with that?

I was noticing that there is no demonstration that the compounds reported exhibit any more activity against MRSA than against any other micro-organisms tested. You focused on MRSA, nothing more, nothing wrong.

“Finally, authors should not overstate their findings, notably in Conclusion lines 394-395: “The outcome of this study demonstrated that at least seven of the isolated compounds from various parts of R. chalepensis possess considerable anti-MRSA property.”

Response: The conclusions are based on facts (findings from the present work). However, to honour the reviewer’s demand, we have now removed the word ‘considerable and replaced with ‘reasonable’. Hope this change will satisfy the reviewer.

Based on facts, simply stating “possess anti-MRSA property” is even better.

“Extraction, isolation and identification of compounds: Detailed spectroscopic data of the isolated compounds are not provided (they could be given as supplementary material). By the way, it is unclear to me whether these compounds were partly or completely reported before, especially in [5]. Please clarify and adapt the manuscript accordingly.

Response: We do not believe that any spectroscopic data of these well known compounds are necessary as supplementary files. That will be too many for 19 compounds!! Besides, appropriate references for published data for all compounds have been provided. That must suffice. The following sentence has been amended to clarify the reviewer’s query. “The isolation and antimicrobial activity of compounds 14-16 against all five microbial strains was reported previously [5].”

According to the section entitled “Correct Identification of Natural Products” in “Instructions for Authors” of the Journal (https://www.mdpi.com/journal/molecules/instructions ): “When previously reported compounds are isolated and used in biological activity assays, the 1H NMR spectrum should be given in the supplementary data as a proof of purity. Authors should consider very carefully potential sources of artifacts and contaminants resulting from extraction procedures or sample handling. When possible compounds should be purified to at least 95% purity. Whatever the claimed purity, it must be fully supported by appropriate analytical techniques and this purity taken into account when reporting and comparing biological activities”.

“The range of concentrations tested for determining MIC is not clearly mentioned.”

Response: I would ask the reviewer to carefully read the resazurin assay as depicted in the paper, and he/she will understand the concentration range based on dilutions.

Please, clearly indicate the maximal and minimal concentrations tested in every case.

“In Table 1, some controls are missing, especially the effect of solvents when used alone.”

Response: Positive controls used were ciprofloxacin against bacterial strains and nystatin against the fungal strain, and the MIC values are shown in Table 2. A footnote has now been added to Table 1 to this effect. There is no solvent effect relevant to this as resazurin microdilution method was used, not disc diffusion or well diffusion.

Ok

“The strains evaluated in Table 1 must be specified (not only mentioning microbial species).”

Response: Not sure what exactly the reviewer is asking for here! The names of all strains tested are present in Table 1.

Please, specify the strain name in each case (e.g. S aureus NCTC…).

“MIC values in Tables 1, 2 and 3 should be provided using the same units (in μg/mL or in μM, as typically used).”

Response: MIC values Tables 1 and 2 are in mg/mL as a standard unit, and that for Table 3 in mg/mL as this is typically used. No need for same units. Both are international units anyway.

“There is no information about technical and biological replicates. Bacteriostatic and/or bactericidal effects should have been determined to better characterize the antimicrobial effects reported.”

Response: Not sure what exactly the reviewer meant by technical and biological replicates! Determination of bactericidal or bacteriostatic effect is not within the scope of this article. We are definitely not performing any extra experiments for this publication, especially in this pandemic situation.

All the data reported in Tables 1, 2 and 3 must be the results of ‘n’ replicates. Please specify the value of ‘n’ in each Table. In a previous paper, you characterized bactericidal and bacteriostatic effects and thus I wondered whether you would have similar data here. This would be a plus for this paper; however, this is obviously only optional, especially if such data are not available at present.

“In silico studies: Many explanations are missing. It looks like this paper was written assuming that readers are familiar with the various bio-informatics analyses conducted, which is not obvious”.

Response: The paper is aimed at specialist readership, not for laymen. However, we have provided all necessary explanations as we felt appropriate. However, if the reviewer would specify any particular explanation, we could have provided that. However, we have now provided additional explanations in some cases.

 “In Materials and Methods, the corresponding section is the shortest.”

Response: Yes, it is not an error! It was kept the shortest intentionally to avoid repetition as much of this info is already present in the Results and Discussion section. We will not make any changes here.

As you want.

“In Table 4, what does mean Pa? Pi? “

Response: Good point. We have now clarified this in Table 4. Pa means probability to be active and Pi means probability to be inactive.

Ok

“In Figures 6 and 7, it is unclear why some items are highlighted (e.g. ORF102, 12 and 35). What is the color code?

Response: All necessary information has now been incorporated in Figures 6 and 7.

 “GI absorption, Lipinski rules and other terms notably used in Table 7 are not defined.”

Response: GI absorption is gastrointestinal absorption. The term is already explained in the text earlier. We have now defined Lipinski rules in the text. The rest of terms are defined as part of the Table legend.

Ok

“Data in Figures 6, 7, 8 and Tables 4, 5, 6 and 7 look only partly exploited. Some of them could be moved to supporting information online, where more explanations would be provided.“

Response: We have now added more description/explanation in the text, and believe that now adequate level of explanation is available. All Figure and Table MUST remain as part of this paper, not as supporting information, for better understanding of the work presented.

As you want

“The MSSA laboratory strain RN4220 is erroneously mentioned once in the Abstract line 27 as a MRSA strain. It is not mentioned thereafter in the manuscript, which suggests it is an error. Please revise.’

Response: Well spotted. We have now deleted RN4220 from the abstract

Ok

“Line 103, Authors state: “Of the 19 isolated compounds from R.chalepensis, 15 compounds were tested […]”, referring then to Table 2. However, this Table summarizes the results of 13 (not15) compounds from R. chalepensis.”

Response: Well spotted again. It was a typo. We have now corrected this to 13.

Ok

“Some typo errors must be corrected. For instances, line 114: “E.Coil”; line 115: “E. Coli”.

Response: All corrected

OK

“Please define all abbreviations used.”

Response: All abbreviations are defined.

OK